# What sustain Chinese adult second language (L2) learners' engagement in online classes? A sequential mix-methods study on the roles of L2 motivation and enjoyment

Yanyan Li[1], Wentao Liu[2]*, Shuai Ren[3]

1 School of Foreign Languages, Changsha University, Changsha, Hunan, China, 2 School of Foreign Studies, Hunan Normal University, Changsha, Hunan, China, 3 School of Politics and Public Administration, South China Normal University, Guangdong, China

* 202310080103@hunnu.edu.cn

## Abstract

Recent research has integrated positive psychology with the Second Language Motivational Self System (L2MMS) to explore how enjoyment, L2 self-guides (including ideal L2 self and ought-to L2 self), and engagement interact among school-aged second-language (L2) learners. However, there is a significant gap in understanding these dynamics among adult learners, particularly those who primarily learn a second language online—a group that has been largely overlooked. To address this gap, our study examined the underlying mechanisms connecting these constructs. We employed a sequential mixed-methods approach with 367 adult L2 learners enrolled in online language courses at three universities in China. Quantitative data were analyzed using structural equation modeling (SEM) with Amos 24, revealing several key findings. Enjoyment was found to directly and positively predict engagement. However, contrary to existing literature, ideal L2 self did not directly predict either enjoyment or engagement. In contrast, ought-to L2 self directly and positively predicted both enjoyment and engagement, and it indirectly influenced engagement through enjoyment. Qualitative data, gathered through semi-structured interviews with five participants and analyzed using MAXQDA 2022, provided deeper insights into these statistical trends. This study concludes by discussing its implications and suggesting directions for future research.

## 1. Introduction

The domain of second language (L2) learning has been profoundly influenced by the tenets of Positive Psychology (PP), which is dedicated to the enhancement of human thriving and well-being [1]. PP has stimulated inquiry into a spectrum of positive factors, including institutional aspects, personal attributes, and emotional states, that are conducive to language proficiency [2]. A pivotal concept within PP, engagement, is particularly instrumental in fostering effective L2 learning [3–5]. Engaged learners are typically associated with superior academic

**Competing interests:** The authors have declared that no competing interests exist.

achievements [6–8], a phenomenon that is highly relevant for L2 learners. In settings where the L2 does not hold official status or widespread usage, learners often encounter a scarcity of genuine opportunities to utilize the L2 in authentic contexts [9, 10]. These learners must be motivated to proactively immerse in the target language and its cultural milieu to cultivate communicative competence [11, 12].

The L2 Motivational Self System (L2MSS) offers a robust framework for deciphering the motivational underpinnings of L2 learners [13, 14]. Central to the L2MSS are L2 self-guides, which encompass ideal L2 self and ought-to L2 self [15, 16]. Recent scholarly work has delved into the dynamics through which these self-guides influence engagement [17–19]. Enjoyment, recognized as a potent positive emotional state by PP [20, 21], has emerged as a significant mediator in this process [22, 23]. More specifically, ideal L2 self has consistently demonstrated a positive impact on enjoyment and engagement, whereas ought-to L2 self's role has not been found to be a significant contributor [22, 24, 25]. These studies are commendable for advancing our comprehension of the motivational and emotional drivers of engagement. However, they are not without limitations, primarily due to their preoccupation with school-aged L2 learners, thereby neglecting the unique context of adult L2 learners.

Adult learners warrant distinctive consideration from researchers, given the complex social and emotional challenges they confront, such as occupational and familial obligations, alongside age-related introspective concerns [26, 27]. These factors are likely to set their motivational and emotional experiences apart from those of their younger peers engaged in L2 learning. Therefore, the generalizability of findings from studies centered on school-aged L2 learners to adult populations may be limited. In light of these considerations, this study employs a sequential mixed-methods approach to re-evaluate the interplay between ideal L2 self, ought-to L2 self, enjoyment, and engagement, specifically within the demographic of Chinese adult L2 learners enrolled in online university courses. The insights yielded from this study have the potential to inform curriculum development, activity planning, and strategic marketing for adult L2 education providers.

## 2. Literature review

### 2.1 Control-value theory as the theoretical basis

We applied the control-value theory (CVT) [28] to address the theoretical connections between L2 self-guides, enjoyment, and engagement. The CVT provides a framework for understanding the antecedents and outcomes of emotions in educational contexts. According to the CVT, control-value appraisals are the proximal antecedent of multiple emotions. Specifically, control appraisal pertains to an individual's perception of their ability to control their achievement activities or outcomes, while value appraisal relates to the perceived significance of these activities or outcomes [29]. The CVT also recognizes distal antecedents, categorized into environmental and personal factors, with motivation being one of the individual factors highlighted by the theory. This suggests that L2 self-guides (ideal L2 self, ought-to L2 self), as motivational elements in the L2 learning context, could elicit emotional responses.

The CVT [28] further suggests that positive emotions like enjoyment, hope, and pride are linked to positive learning outcomes such as increased engagement and improved academic performance [30–32]. Conversely, negative emotions such as anxiety, boredom, and anger may lead to adverse learning outcomes. From this, we can deduce that enjoyment could lead to greater engagement. Additionally, the CVT outlines cognitive-motivational mechanisms that influence learning outcomes through a series of cognitive and motivational factors [33, 34]. In the L2 learning context, L2 self-guides represent a motivational factor that could impact engagement. Thus, we can infer that L2 self-guides may have a direct influence on the level of

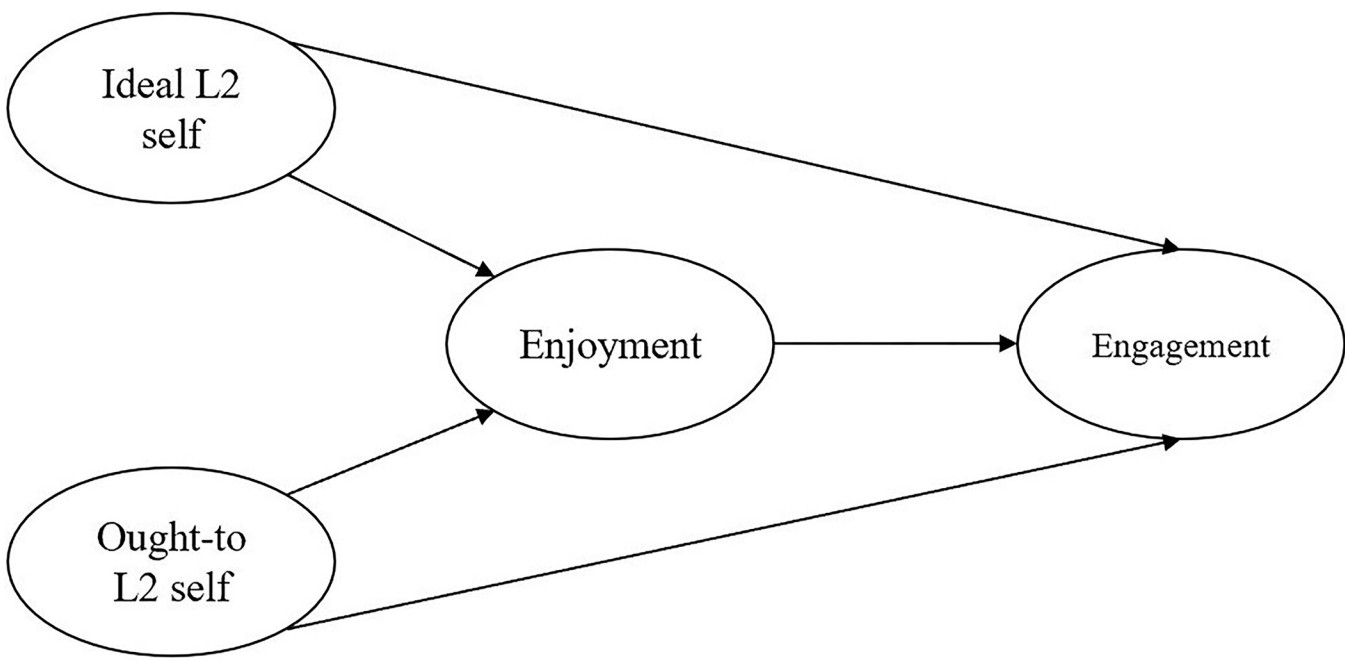

**Fig 1. Theoretical links between L2 self-guides (ideal L2 self, ought-to L2 self), enjoyment, and engagement.**

engagement in language learning activities. The theoretical links between these variables were visualized in Fig 1.

## 2.2 Engagement in L2 contexts

Over the past two decades, the concept of student engagement has garnered significant scholarly interest [4, 35, 36]. This construct encapsulates the proactive involvement of students in educational activities and their dedication to achieving educational objectives [4, 37]. There is a well-established correlation between student engagement and academic performance [38, 39]. In the L2 learning context, student engagement is commonly delineated into three interconnected dimensions: behavioral, emotional, and cognitive [40, 41]. Behavioral engagement pertains to the extent and nature of students' active involvement in language acquisition activities, encompassing adherence to classroom protocols, voluntary participation, proactive interaction, and perseverance in completing tasks [42]. Emotional engagement reflects the affective responses of learners, characterized by positive attitudes towards learning content, educational activities, and interpersonal dynamics within the language learning milieu [9, 43]. Cognitive engagement, on the other hand, denotes the intellectual effort and strategic processes that students employ during language learning. This involves the intentional, selective, and enduring allocation of attention to synthesize various learning resources in the pursuit of educational goals [44].

## 2.3 L2 self-guides

Motivation is a pivotal determinant of success in L2 learning endeavors. While Gardner (45)'s integrative motivation has historically been influential, its explanatory power regarding motivated behavior is not universally consistent across different cultural landscapes [13]. In response to this limitation, Dörnyei [45] introduced an innovative motivational framework known as the L2 Motivational Self System (L2MSS). This framework synthesizes the socio-

educational model [46], possible selves theory [47], and self-discrepancy theory [48] to offer a more nuanced perspective on motivation in language learning contexts. At the heart of the L2 L2MSS are the concepts of 'ideal L2 self' and 'ought-to L2 self,' which have attracted significant academic interest [49, 50]. These concepts are collectively referred to as "L2 self-guides" [51]. Ideal L2 self encapsulates the motivational drive stemming from an individual's aspiration to reduce the gap between their current linguistic proficiency and the envisioned ideal level of competence they aspire to attain [52]. Ought-to L2 self represents the motivational force derived from the fulfillment of expectations imposed by significant others, such as educators, peers, and family members, or from the desire to evade adverse outcomes [53].

## 2.4 Enjoyment in L2 contexts

Enjoyment stands as a quintessential and pervasive positive emotion encountered by L2 learners [7, 54, 55]. This construct can be delineated as the perception of novelty or a feeling of accomplishment that arises when individuals surpass their own expectations and attain previously unforeseen objectives [56]. Enjoyment is predominantly experienced in the pursuit and dedication to goals that hold personal significance [57]. Empirical evidence has demonstrated that enjoyment significantly enhances L2 learners' willingness to communicate[3, 23, 58]. Furthermore, enjoyment has been linked to positive outcomes in terms of L2 achievement [20, 22, 51]. The significance of enjoyment in L2 learning is fundamentally underpinned by Fredrickson's (2003) Broaden-and-Build theory, a cornerstone of Positive Psychology. According to Fredrickson [59] positive emotions have the capacity to expand individuals' momentary cognitive and behavioral repertoires, accumulate personal resources, and foster resilience amidst adversity. This elucidates the potent influence of enjoyment on the language learning experience [1], suggesting that positive emotional states can create an enabling environment for linguistic growth and communicative engagement.

## 2.5 Empirical links between L2 self-guides, enjoyment, and engagement in online contexts

Up until now, research on the link between self-guides and engagement in online learning contexts has been scarce. However, a substantial body of work in traditional, face-to-face settings has demonstrated the impact of self-guides on engagement. Notably, ideal L2 self has been repeatedly recognized as a key driver. For instance, in the broader field of L2 education, Sun, Shi [18] investigated how ideal L2 self influences engagement among 466 L2 learners. Their findings indicated that ideal L2 self directly and positively influences learner engagement. The significance of ideal L2 self has also been noted in specific language skills, such as reading. Abdollahzadeh, Amini Farsani [24] discovered that this construct is a powerful predictor of engagement in academic L2 reading among 419 university students. Echoing this, Tsao, Tseng [25] found that ideal L2 self significantly boosts engagement in L2 writing courses among 433 university students. Conversely, ought-to L2 self has generally been seen to play a non-significant role in boosting learner engagement, as confirmed by researchers including Abdollahzadeh, Amini Farsani [24], Sadoughi, Hejazi [60], and Tsao, Tseng [25].

A review of the extant literature indicates a scarcity of studies that delve into the connection between self-guides and the enjoyment derived from online learning experiences. In contrast, a wealth of research in traditional classroom settings has examined the role of self-guides in affecting enjoyment. Ideal L2 self has notably been identified as a potent motivator. For instance, the study of Fathi and Hejazi [23], which involved 452 college students, demonstrated that this construct can initially boost enjoyment and subsequently lead to improved language proficiency. The impact of ideal L2 self is not confined to general proficiency but extends to

specific language skills such as writing. Zhao, Zhu [61], in their research involving 239 college students, confirmed its positive influence on enjoyment. Longitudinal research further supports this, as evidenced by the study of Fathi, Pawlak [62] which, using a cross-lagged panel design with 903 college students, uncovered a significant lagged effect of ideal L2 self on enjoyment. Conversely, ought-to L2 self appears to have a negligible impact on enhancing learner enjoyment, a finding corroborated by various researchers, including Tahmouresi and Papi [22], Liu, Wang [51], and Kim [63].

Additionally, the relationship between enjoyment and engagement in online learning contexts has been well-documented, as evidenced by research conducted by Wang and Hui [64], Dai and Wang [54], Derakhshan and Fathi [65], which primarily involved school-aged students. The above review of the interaction between self-guides, enjoyment, and engagement suggests that while the influence of enjoyment on engagement is acknowledged in online L2 learning, the impact of self-guides on both engagement and enjoyment has not been thoroughly explored in these settings. Existing studies have highlighted differences in emotional experiences and outcomes among L2 learners in online versus offline contexts, as noted by Li, Guan [66] and Resnik, Dewaele [67]. This underscores the necessity for a study that reevaluates these relationships specifically in online L2 learning contexts. Furthermore, the existing body of literature primarily targets school-aged L2 learners, neglecting the unique needs of adult learners. This oversight is significant given that adult learners often juggle language learning with other professional and personal responsibilities [27]. Therefore, the inclusion of adult learners in motivational research is not only crucial but also long overdue, considering their distinct sociocultural and educational backgrounds compared to their younger counterparts.

## 3. Research hypotheses

Based on the above review on both theoretical and empirical literature pertaining to self-guides (ideal L2 self, ought-to L2 self), enjoyment, and engagement. We formulated the following hypotheses and aimed to test them among Chinese adult L2 leaners in online contexts:

**H1:** Ideal L2 self could positively and significantly predict engagement.

**H2:** Ideal L2 self could positively and significantly predict enjoyment.

**H3:** Enjoyment could positively and significantly predict engagement.

**H4:** Ideal L2 self could positively and significantly predict engagement via enjoyment.

**H5:** Ought-to L2 self could not significantly predict engagement.

**H6:** Ought-to L2 self could not significantly predict enjoyment.

**H7:** Ought-to L2 self could not significantly predict engagement via enjoyment

## 4. Methodology

### 4.1 Participants

For quantitative phase, this study utilized purposive sampling to select 388 adult L2 learners enrolled in the online adult L2 training programs provided by a university in China. They were from diverse industries and had different majors. After excluding 21 participants who either gave identical responses to all items or submitted incomplete responses, the final sample consisted of 367 participants from diverse industries (male: 147, female: 220; age: 27–62 years old, M = 33.41). As for qualitative phase, to avoid bias, we also randomly invited 5 participants

**Table 1. Demographic information of interviewees.**

| Interviewee | Gender | Age | Job | L2 |
|---|---|---|---|---|
| 1 | Female | 32 | Public servant | English |
| 2 | Male | 30 | Civil engineer | English |
| 3 | Female | 36 | International trade salesperson | English |
| 4 | Female | 29 | Clerk | French |
| 5 | Male | 31 | Software developer | English |

with different jobs to have a semi-structured interview. Their detailed demographic information is presented in Table 1.

To ensure the validity of the self-reported scales on self-guides, enjoyment, and engagement, we employed a translation and back-translation procedure. Three bilingual researchers translated the scales from English to Chinese and then back-translated them into English. Subsequently, an expert in psycholinguistics and a translation expert reviewed and revised the wording to ensure the closest semantic equivalence between the English and Chinese versions. Participants were required to complete the Chinese scales, with the option to reference the original English scales for clarification as needed.

## 4.2 Instruments

We would like to clarify that the adaptation of the scale items was intended to make them more relevant for our participants, who are adult L2 learners. The original scale items were designed for school students, which prompted us to modify them accordingly. Additionally, the original scales assessing enjoyment and engagement were developed for offline learning contexts, so we tailored the items to better fit a career-focused and online learning environment. It is important to note that no new items were created in the process.

The entire adaptation was carried out collaboratively by both authors, through careful discussion and negotiation. To further ensure the relevance of the adapted items, two adult L2 learners were involved in the process to confirm that the items accurately reflected their learning experiences. However, to maintain impartiality, these two adult L2 learners were not included as participants in the study.

**4.2.1 Scale for assessing L2 self-guides.** The self-guides measures were adapted from Li [68]. To better fit adult L2 learners, we adjusted the items of ought-to L2 self by replacing "family members" with "my employers" or similar terms. Sample items from each dimension included statements such as 'I can imagine myself frequently speaking a L2 with international friends' and 'I consider learning a L2 important because my employers think that I should do it'. Responses were rated on a 7-point Likert scale, ranging from 1 (strongly disagree) to 7 (strongly agree). All items were positively framed, with higher scores indicating stronger levels of ideal L2 self or ought-to L2 self. The scale demonstrated good reliability in our study, with Cronbach's α of 0.803 for ideal L2 self and 0.919 for ought-to L2 self.

**4.2.2 Scale for assessing enjoyment.** We utilized the short version of Achievement Emotions Questionnaire (AEQ) class-related emotion subscales to assess participants' positive emotions, as each specific subscale can function independently [69]. To better align with online L2 learning contexts, we refined the wording of items in the enjoyment subscale, which comprised 10 items. For example, a sample item was "I enjoy being in online L2 class." Participants rated their feelings about L2 learning on a 7-point Likert scale ranging from 1 (strongly disagree) to 7 (strongly agree). The scale exhibited strong reliability in our study, with a Cronbach's α of 0.968.

**4.2.3 Scale for assessing engagement.**   We employed the short Language Classroom Engagement Scale (LCES) to gauge participants' online engagement in the L2 contexts [40]. Adaptations were made to ensure its relevance to online settings. The LCES comprises three subscales: behavioral engagement (3 items; e.g., "I put effort into learning in my online L2 class."), emotional engagement (3 items; e.g., "I look forward to my online L2 class."), and cognitive engagement (3 items; e.g., "In my online L2 class, I think about different ways to solve a problem."). Respondents rated each item on a 7-point Likert scale ranging from 1 (never) to 7 (always), where higher scores indicate greater levels of online engagement. The scale showed great reliability in our study, with a Cronbach's α of 0.975.

**4.2.4 Sem-structured interview.**   We conducted semi-structured interviews with 6 participants who had previously completed our questionnaire survey. Each interview lasted approximately 20 minutes. The primary aims were twofold: firstly, to deepen our understanding of how motivational factors identified in the quantitative stage influence online engagement; and secondly, to elucidate the underlying reasons behind the quantitative findings. To achieve these exploratory objectives, we opted for an inductive semi-structured interview approach. The interview questions were meticulously crafted following input from three psycholinguistics experts and a pilot test involving two students:

*RQ1*: Do you like the L2 you are learning in the online L2 class?

*RQ2*: What factors motivate you to be engaged in the online L2 class? Could you describe them in detail.

## 4.3 Data collection procedure

Our study collected quantitative data during the summer in 2023. The data was gathered from three universities in China that offer multilingual online training for adult L2 learners. These courses encompass languages such as English, Russian, French, Korean, Japanese, etc., catering to learners at beginner, intermediate, and advanced levels. Courses are delivered via live broadcasting. The curriculum is designed to incorporate phonetics, vocabulary, and grammar tailored to each proficiency level, providing a structured learning trajectory aimed at enhancing language proficiency progressively. This approach is intended to meet the diverse educational, vocational, and examination requirements of adult L2 learners.

Our collaborative partnerships with university administrators facilitated data collection efforts. Subsequently, L2 instructors distributed an online questionnaire link through a WeChat group chat during scheduled breaks, accompanied by a thorough explanation of the study's objectives, procedures, and confidentiality. Participation in the survey was voluntary, with these leaners incentivized by receiving a 20 RMB WeChat red packet upon completion.

Qualitative data collection involved random sampling of five participants per university due to our finical constraints, selected by L2 instructors, who agreed to participate in subsequent interviews. Each participant received a 500 RMB WeChat red packet. Prior to interviews, informed consent was obtained from all participants. This study was conducted from October 15, 2023, to September 22, 2024. It received ethical approval from the Ethics Committee of Hunan Normal University (No. 2023673) and followed the principles set forth in the Helsinki Declaration.

## 4.4 Data analysis

This study employed AMOS 24 to perform quantitative data analysis. Following the two-stage structural equation modeling (SEM) approach suggested by Kline [70], the measurement model was verified first, which assessed the reliability and validity of the constructs. Several

indices were used to evaluate the goodness-of-fit of the models, including Chi-square divided by degree of freedom ($\chi^2$/df), Comparative Fit Index (CFI), Tucker–Lewis Index (TLI), and Root Mean Square Error of Approximation (RMSEA) [71]. Next, the data were examined for skewness, kurtosis, and descriptive statistics to ensure their normality and suitability for further analysis. Then, the structural model was tested, which examined the causal relationships among the constructs. The "Maximum Likelihood" method was used to estimate the parameters of the models. Moreover, the bootstrapping technique with 5000 iterations was applied to test the significance of the indirect effects [72]. The bootstrapping analysis computed the upper and lower limit confidence intervals. The indirect effects were considered significant if zero was not included in the 95% confidence intervals (CI) of the bootstrapping analysis [73].

The interviews, originally conducted in Chinese, were transcribed by us. With the assistance of two linguistics professors, we were able to translate the text into English. Utilizing MAXQDA 2022, we applied thematic analysis in six phases as outlined by Braun and Clarke [74]: familiarization, initial coding, searching for themes, reviewing themes, defining themes, and reporting. Since the goal of the qualitative phase is to identify the motivational factors driving Chinese adult L2 learners' language learning, providing insights for the quantitative findings, the extracted themes are categorized as either "ideal L2 self" or "ought-to L2 self". Our qualitative analysis followed an inductive approach, allowing themes to emerge naturally from the data. To enhance the repeatability and transparency of our coding process (see Table 2), we include an example of an interview excerpt from the results section as follows:

> *"I'm focused on accessing information, crucial for my computer science background. I've come to understand the significance of English in my career, as <u>programming languages and tools predominantly originate from Western countries</u>. When facing technical challenges, <u>the most reliable solutions and cutting-edge technologies are often documented in English</u>. Relying on translations can result in inaccuracies and additional costs, whereas English resources are freely available. <u>To bridge the information gap and stay competitive</u>, mastering English is indispensable."*

To ensure reliability, the first author and corresponding author independently coded the data, achieving agreement on 61 out of 65 codes (93.85%) after resolution of disagreements

**Table 2. The example of the detailed coding process.**

| Procedures | Explanation |
| --- | --- |
| 1. Familiarization | The focus is on career advancement, access to information, and staying competitive in a technical field. There is a sense of practical necessity and career-related motivation. |
| 2. Initial coding | **Code 1**: "*Programming languages and tools from Western countries*"–English is essential for technical resources.<br>**Code 2**: "*Reliable solutions and cutting-edge technologies are documented in English*"–English is key for accessing up-to-date, accurate resources.<br>**Code 3**: "*Bridge the information gap and stay competitive*"–Motivation stems from the need to stay competitive by overcoming knowledge gaps. |
| 3. Searching for themes | **"Accessing information"** –The participant emphasizes the importance of English for accessing crucial resources based on codes 1, 2, and 3. |
| 4. Reviewing Themes | **"Accessing information"** is driven by the professional need to master English, making it a clear example of a "**Career-driven necessity**". |
| 5. Defining Themes | Check if the themes align with the data and fit within the broader motivational framework (ideal vs. ought-to L2 self). Clearly, "**Career-driven necessity**" aligns with **ought-to L2 self.** |
| 6. Reporting | The interview data reveals a clear focus on career-driven necessity, namely, **ought-to L2 self**. |

through discussion. Multiple strategies were employed to enhance the validity of our qualitative findings. Member checking involved inviting interviewees to review and validate extracted codes/themes and interpretations, thereby bolstering credibility [75]. Additionally, providing a detailed description of the research context, interviewees, and data collection procedures enhanced the dependability of our findings. Bracketing and audit trialing were used to prevent bias and ensure transparency in the data analysis process. By rigorously applying these strategies, we upheld criteria for trustworthiness, resulting in valid and reliable findings [76].

## 5. Quantitative results

### 5.1 Testing the measurement model

We performed confirmatory factor analysis (CFA) to ensure construct validity. The initial model consisted of four constructs: ideal L2 self, ought-to L2 self, enjoyment, and engagement. Among them, engagement had three components, making the model a second-order one. The model was tested for non-significant or low factor loadings using both unstandardized and standardized estimates. Items OL2 were removed as their factor loadings were below 0.45. Next, CFA was conducted again. Fig 2 presents the final CFA results.

The measurement model's goodness of fit was tested using the criteria suggested by Hu and Bentler (1999). The CFA results indicate that the measurement model fitted the data well, as shown by the following indices: $\chi^2/df$ = 2.719 (3–5: acceptable, < 3: excellent), CFI = 0.963 (> 0.9: acceptable, > 0.95: excellent), TLI = 0.957 (> 0.9: acceptable, > 0.95: excellent), RMSEA = 0.069 (< 0.08: acceptable), and SRMR = 0.043 (< 0.08: acceptable).

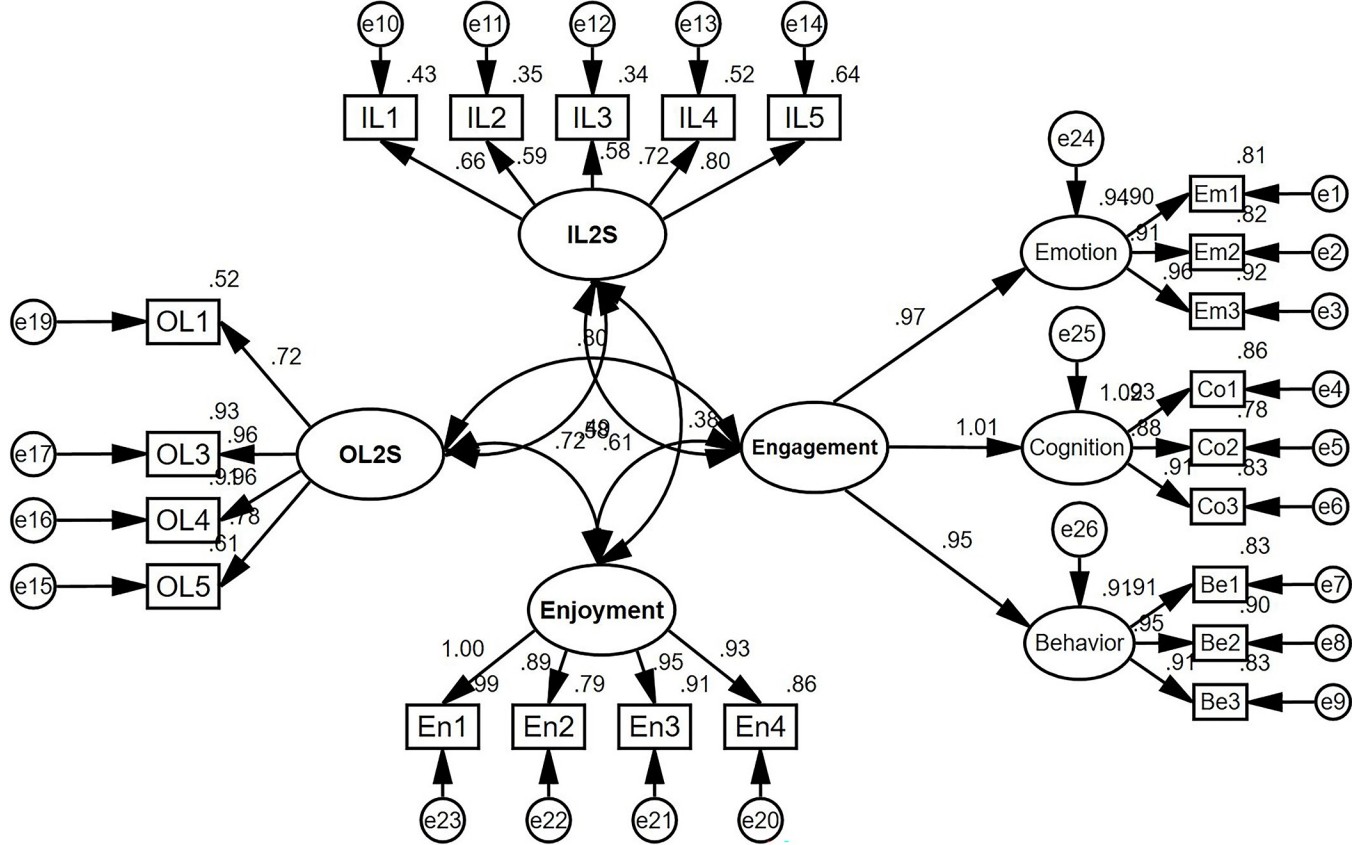

**Fig 2. The final modified measurement model ($n$ = 367).**

**Table 3. Convergent and discriminant validity of the measurement model (*n* = 367).**

| Construct | CR | AVE | MSV | Fornell-Larcker Criterion | | | |
|---|---|---|---|---|---|---|---|
| | | | | 1 | 2 | 3 | 4 |
| 1. Ideal L2 self | 0.806 | 0.557 | 0.338 | **0.746** | | | |
| 2. Ought-to L2 self | 0.920 | 0.745 | 0.642 | 0.582*** | **0.863** | | |
| 3. Enjoyment | 0.969 | 0.887 | 0.515 | 0.382*** | 0.718*** | **0.942** | |
| 4. Engagement | 0.985 | 0.957 | 0.642 | 0.489*** | 0.801*** | 0.611*** | **0.978** |

*Note.* CR = Composite Reliability; AVE = Average Variance Extracted; MSV = Maximum Shared Variance. Figures in bold show the square root of AVE for each construct.

According to Table 3, All constructs met the criteria of Composite Reliability (CR) > 0.7 and Average Variance Extracted (AVE) > 0.5, and had Maximum Shared Variance (MSV) values lower than their AVE, demonstrating convergent validity [77]. The Fornell-Larcker criterion further indicated that all factors were interrelated. The discriminant validity was verified by the fact that the square root of AVE for each construct (the bold values in Table 3) exceeded its correlations with other factors [77].

## 5.2 Testing the structural model

After verifying the construct validity, convergent validity, and discriminant validity of the measurement model, we used the data in the model for further analysis. Table 4 presents the descriptive statistics of the data. It shows that the scores of each construct involved in this study were normally distributed, as their skewness and kurtosis values were below the absolute value of 2. Hence, the data met the assumptions of the SEM.

Pearson correlation analysis was employed to examine the relationships among the constructs. The correlation coefficients are presented in Table 5. As indicated in Table 5, all constructs exhibited significant correlations, indicating that the data fulfilled the assumptions of the SEM.

To investigate how the ideal L2 self, ought-to L2 self, enjoyment, and engagement in online classes interrelate among Chinese adult L2 learners, we conducted a regression analysis employing SEM. The structural model showed excellent model fit, as shown by the following indices: $\chi^2$/df = 1.623, CFI = 0.998, TLI = 0.995, RMSEA = 0.041, and SRMR = 0.010. The structural model in this analysis is depicted in Fig 3.

Based on Fig 3, ideal L2 self did not predict engagement ($\beta$ = 0.008, $p$ > 0.05) and enjoyment ($\beta$ = -0.054, $p$ > 0.05) significantly, thereby rejecting H1 and H2. Besides, enjoyment positively and significantly predict engagement ($\beta$ = 0.113, $p$ < 0.05), thus supporting H3.

**Table 4. Descriptive statistics of all constructs (*n* = 367).**

| Construct | Min | Max | M | SD | Skewness | Kurtosis |
|---|---|---|---|---|---|---|
| Ideal L2 self | 1 | 7 | 4.349 | 1.316 | -0.716 | 0.428 |
| Ought-to L2 self | 1 | 7 | 4.088 | 1.647 | -0.262 | -0.643 |
| Enjoyment | 1 | 7 | 3.359 | 1.644 | 0.375 | -0.596 |
| Emotion | 1 | 6 | 3.621 | 1.366 | -0.385 | -0.420 |
| Cognition | 1 | 6 | 3.528 | 1.329 | -0.274 | -0.337 |
| Behavior | 1 | 6 | 3.448 | 1.355 | -0.184 | -0.496 |

*Note.* Emotion, cognition, and behavior are the components of engagement; M = mean; SD = standard deviation.

**Table 5. Correlations between all constructs ($n = 367$).**

| Constructs | 1 | 2 | 3 | 4 | 5 | 6 |
|---|---|---|---|---|---|---|
| OL2S | - | | | | | |
| IL2S | 0.542** | - | | | | |
| Enjoyment | 0.669** | 0.325** | - | | | |
| Emotion | 0.759** | 0.421** | 0.570** | - | | |
| Cognition | 0.766** | 0.414** | 0.574** | 0.913** | - | |
| Behavior | 0.713** | 0.390** | 0.544** | 0.866** | 0.915** | - |

*Note.* Emotion, cognition, and behavior are the components of engagement

**$p < 0.001$.

Additionally, ought-to L2 self significantly and positively predicted engagement ($\beta = 0.704$, $p < 0.05$) and enjoyment ($\beta = 0.698$, $p < 0.05$), thereby corroborating H5 and H6.

Table 6 presents the findings from the indirect path analysis. It was observed that the ideal L2 self did not significantly predict engagement through enjoyment ($\beta = -0.006$, 95% CI [-0.020, 0.000]), which led to the rejection of H4. Conversely, Ought-to L2 self was found to significantly predict engagement through enjoyment ($\beta = 0.079$, 95% CI [-0.015, 0.134]), thereby rejecting H7.

## 6. Qualitative results

The interview data were analyzed to investigate the reasons for rejecting H1, H2, and H4. For interview RQ1 (Do you like the L2 you are learning in the online L2 class?), only one of the five interviewees confirmed that they liked the L2 they were learning in the online class. For the motivations (interview RQ2), 65 codes, 10 categories, 3 sub-themes, and 2 themes were extracted. The thematic map is presented in Fig 4.

Fig 4 depicts the primary themes as follows: the ought-to L2 self (60 codes) was predominant. Within this category, there were two subthemes: educational pursuit (21 codes) and career development (39 codes), each encompassing several categories. Conversely, the ideal L2 self (5 codes) featured a single sub-theme—personal experience (5 codes)—which comprised just two categories.

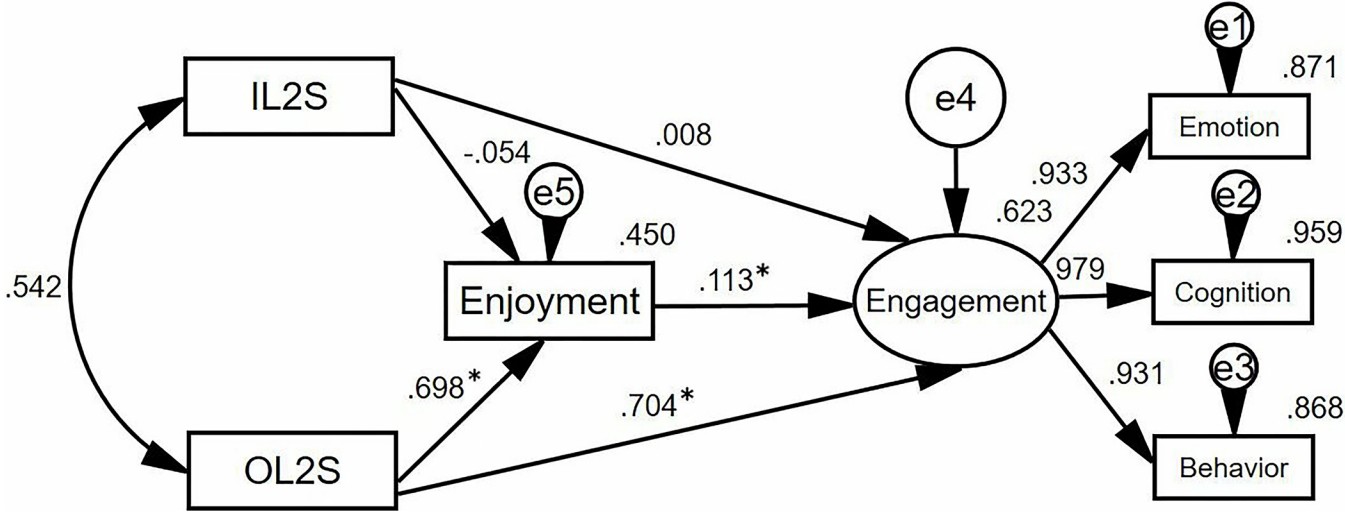

**Fig 3. The structured model ($n = 367$).** *Note.* *$p < 0.05$; IL2S = ideal L2 self; OL2S = ought-to L2 self.

**Table 6. The results of indirect path analysis (*n* = 367).**

| Indirect path | β | SE | 95% CI | *p* |
|---|---|---|---|---|
| 1. Ideal L2 self → Enjoyment→ Engagement | -0.006 | 0.006 | [-0.020,0.000] | 0.102 |
| 2. Ought-to L2 self → Enjoyment→ Engagement | 0.079 | 0.037 | [0.015,0.134] | 0.037 |

*Note*: β = standardized coefficient; SE = standard error.

We provide one interview example per sub-theme, with underlined text indicating the key words used for category extraction. Regarding the ought-to L2 self, career development emerges as a primary motivator for Chinese adults' L2 learning. For instance, Interviewee 5 illustrated how proficiency in English enables individuals to gain firsthand information, enhancing competitiveness in their careers:

> *I'm focused on accessing information, crucial for my computer science background. I've come to understand the significance of English in my career, as programming languages and tools predominantly originate from Western countries. When facing technical challenges, the most reliable solutions and cutting-edge technologies are often documented in English. Relying on translations can result in inaccuracies and additional costs, whereas English resources are freely available. To bridge the information gap and stay competitive, mastering English is indispensable.——Interviewee 5 (Software developer)*

On the other hand, Interviewee 1 described the connection between L2 proficiency and educational pursuit, another sub-theme of ought-to L2 self,

> *I studied English with the goal of future educational advancement. Initially, I, along with others in the government sector, doubted the practicality of learning English due to its limited*

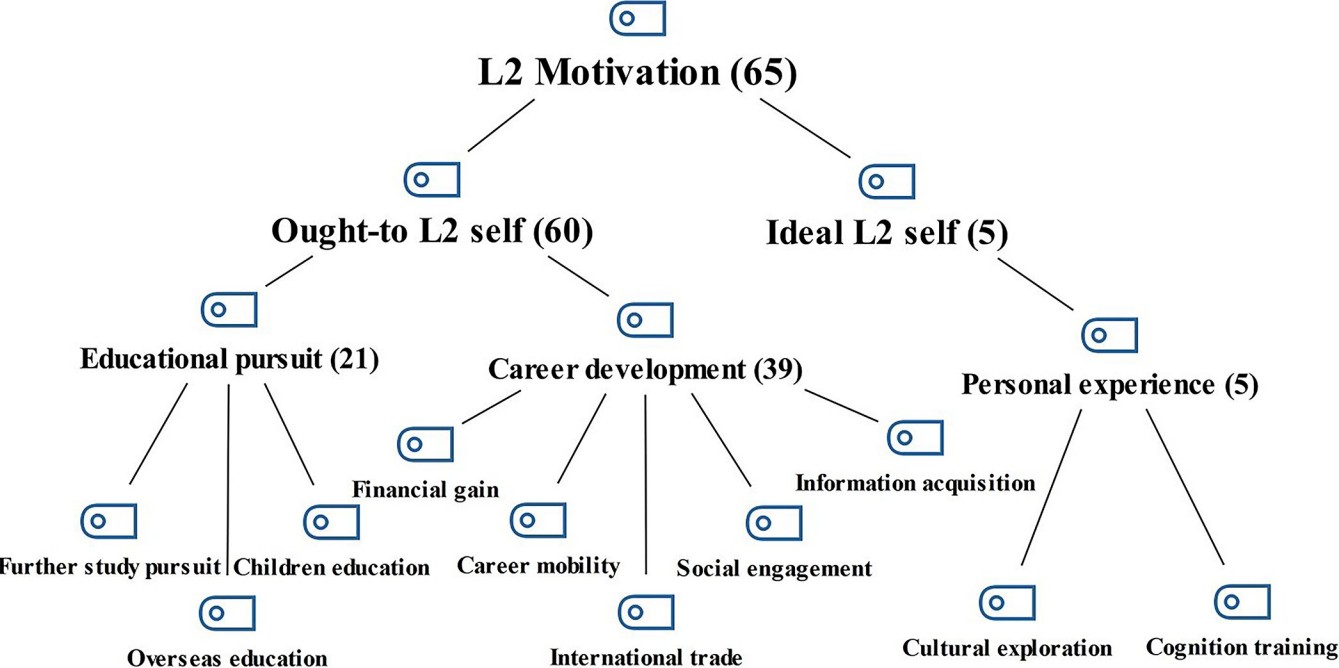

**Fig 4. Thematic map of Chinese adult L2 leaners' motivation for attending online L2 class.**

*use. However, my seven years of experience there revealed the unexpected advantages of English proficiency. For anyone eyeing an MPA or part-time graduate studies, or a full-time academic return, <u>English is an essential exam component.</u> I've noticed many colleagues struggle with English exams due to post-work language disuse, which hampers their preparation. Post-admission, the stringent demands of academic and master's English courses add to the challenge.——Interviewee 1 (Public servant)*

Regarding their ideal L2 self, many interviewees expressed a desire to experience Western cultures independently, each with varying motivations. Interviewee 2, for instance, stated:

*My motivation for learning English is not driven by utilitarianism, such as finding a good job or obtaining a higher degree. During my school years, I pondered why ancient China, once so powerful, fell behind Western countries in modern times. I didn't want to find the answer in textbooks. I wanted to visit those countries to understand <u>the cultures of the Western powers.</u> With English as the lingua franca, <u>mastering it is incredibly important.</u>——Interviewee 2 (Civil engineer)*

## 7. Discussion

Drawing on the CVT, we employed a mixed-method approach to examine the mechanisms underlying the relationship between L2 self guides (ideal L2 self and ought-to L2 self), enjoyment, and engagement in online class among Chinese adult L2 learners, the demographics that received limited attention by L2 researchers. In the quantitative analysis, we first verified the reliability and validity of the instruments used in the study. Afterward, we conducted SEM to examine the proposed hypotheses. To have an insight into these statistical patterns, semi-structured interviews were subsequently conducted.

Our quantitative data derived failed to substantiate Hypotheses 1, 2, and 4, which had posited direct associations between the ideal L2 self, enjoyment, and engagement. Our analysis revealed that among Chinese adults enrolled in online L2 courses, ideal L2 self did not significantly predict engagement or enjoyment experienced. This outcome implies that the learners' conceptualizations of an optimal L2 self-image did not exert a discernible impact on their active participation in the linguistic educational endeavor. Moreover, ideal L2 self did not contribute to a heightened sense of pleasure or fulfillment in the learning milieu for this cohort. The absence of a mediating effect of enjoyment on the relationship between the ideal L2 self and engagement within the L2 learning process was also observed. These findings challenge the conventional assumptions of the CVT [28], which posits that motivational forces can engender positive emotions, thereby fostering increased engagement in activities. Our results stand in stark contrast to the prevailing literature on school L2 leaners and L2 learning contexts outside of the online domain, where the ideal L2 self has been consistently identified as a robust predictor of learner engagement [18, 24, 25] and affective responses such as enjoyment [23, 23, 61].

The divergence of our findings from the established body of work warrants further exploration. Subsequent qualitative analyses elucidate potential reasons behind this discrepancy. Among Chinese adult L2 learners in our study, the aspiration to become a proficient L2 speaker capable of engaging in fluent cross-cultural communication—emblematic of the 'ideal L2 self'—did not emerge as a principal motivation for language learning. A solitary participant cited exposure to diverse cultures as the impetus for learning an L2, while the remaining participants either omitted this goal or alluded to it only in passing. This apparent lack of a strong

ideal L2 self among the Chinese adult learners in our study may underpin the observed lack of statistical significance in its predictive capacity for both enjoyment and engagement in online L2 learning contexts. The qualitative data, therefore, offer a compelling explanation for the quantitative findings, suggesting that the 'ideal L2 self' may not serve as a universal motivational construct across different learner populations and learning modalities, particularly the former.

Furthermore, our study's findings revealed the positive and significant predictive effect of enjoyment on engagement in online L2 classes, thereby affirming Hypothesis 3. This result indicates that among the Chinese adult learners in our sample, heightened affective enjoyment was a precursor to increased engagement in the online L2 learning context. The observed positive correlation between enjoyment and engagement supports the notion that a more pleasurable learning experience can foster a deeper level of commitment and participation in educational activities. Our findings are consistent with recent studies by Dai and Wang [54], Derakhshan and Fathi [65], and Wang and Hui [64], which have reported similar positive associations between enjoyment and engagement in the context of online L2 learning. These collective results reinforce the validity of the CVT 's proposition that positive emotions, such as enjoyment, can lead to increased motivation and engagement in academic pursuits [30]. The affirmation of Hypothesis 3 within our study contributes to the growing body of evidence that underscores the importance of affective factors in online L2 learning. It highlights the need for educational practitioners and researchers to consider the role of enjoyment in enhancing learner engagement and suggests that strategies aimed at increasing positive affective experiences may be beneficial in optimizing the online L2 learning process.

Our quantitative analysis further found that ought-to L2 self positively and significantly predicted both engagement and enjoyment in online classes. This direct effect supports Hypotheses 5 and 6, suggesting that the sense of necessity associated with L2 learning can enhance the affective and behavioral outcomes. Moreover, ought-to L2 self was revealed to exert a positive and significant indirect effect on engagement through the mediation of enjoyment, thus corroborating Hypothesis 7. This implies that the obligatory aspects of L2 learning can influence engagement not only directly but also indirectly by enhancing the learners' affective experience. These results introduce an unexpected nuance to the existing literature, which has predominantly reported a weaker predictive power of ought-to L2 self on engagement and enjoyment among school-aged students [64, 65, 67]. Subsequent qualitative inquiries shed light on potential explanations for this discrepancy. A majority of interviewees associated L2 learning with career advancement, a notion that encapsulates ought-to L2 self. They detailed the direct and indirect benefits of L2 proficiency, such as improved access to information, social networking, international trade facilitation, enhanced employability, financial rewards through language teaching, and the pursuit of higher education domestically and abroad. Additionally, they cited the ability to provide better educational opportunities for their children as an indirect benefit. The qualitative narratives substantiate our statistical findings.

## 8. Theoretical and practical implications

This study significantly contributes to both theoretical and practical aspects of L2 education, particularly focusing on adult learners. Theoretically, it fills a notable gap in L2 literature by specifically addressing adult L2 learners, a demographic historically underrepresented in scholarly research compared to younger, school-aged learners. Furthermore, our findings advance theoretical discourse by showing that among adult L2 learners, ought-to L2 self—rather than ideal L2 self—emerges as a more influential predictor of enjoyment and engagement. This discovery places the CVT framework within a more nuanced context, affirming its

relevance to the distinct motivational and emotional dynamics observed in this demographic. Additionally, this study, to our best knowledge, is the first one that explores specific motivational aspects among Chinese adult L2 learners. These findings open new avenues for understanding the complex nature of motivation in language learning.

From a practical standpoint, to effectively leverage the concept of ought-to L2 self in online language programs, course designers and instructors need to consider the unique motivations of adult learners. These learners often have specific, pragmatic reasons for learning a second language (such as career advancement, travel, or meeting professional obligations), so creating an environment that aligns with these motivations is crucial for enhancing their enjoyment and engagement. Here are some specific strategies:

1. *Incorporating real-world application and career-relevant content*
   To enhance learner motivation, course content should be directly tied to real-world scenarios or professional contexts, highlighting how language proficiency can improve job performance and career opportunities. This can be achieved by incorporating case studies, role-playing exercises, industry-specific vocabulary, and problem-solving tasks that reflect the learner's work environment. For instance, an online English program tailored for healthcare professionals might include medical terminology, communication strategies for patient interaction, and simulations of real-life situations. By aligning course materials with learners' professional responsibilities or career goals, instructors help them visualize how language learning connects to their ought-to L2 self—the expectations they feel from work, family, or society. This alignment makes the learning process more personally relevant and motivating, as learners see a clear link between their language skills and their broader life objectives.

2. *Setting clear and achievable goals with progress tracking*
   Establishing clear, measurable learning goals is essential for helping learners track their progress in meaningful ways. These goals should align with learners' external motivations, enabling them to visualize the steps needed to achieve their professional or personal aspirations. To support this, tools such as progress dashboards, badges, or certificates can be used to mark milestones, while encouraging learners to set specific personal language goals—such as mastering business-related phrases, passing a certification exam, or improving communication with colleagues. Celebrating these achievements reinforces their sense of accomplishment. Adult learners are often motivated by tangible outcomes, and setting clear goals not only provides a roadmap for success but also helps them feel accountable to both themselves and the external expectations tied to their careers or job requirements.

3. *Fostering social interaction and community building*
   Creating opportunities for learners to engage in real-life communication is a powerful strategy, especially for adult learners who view language as a tool for social or professional networking. This can be achieved by facilitating group discussions, peer reviews, collaborative projects, or live virtual sessions where learners participate in conversations or debates relevant to their field of work. Additionally, learners can be encouraged to form study groups or discussion forums where they can share professional experiences and language-learning tips. Social interaction not only reinforces the idea that language is a means to connect with others—such as colleagues, clients, or the broader professional community—but also helps learners build their ought-to L2 self, motivating them to meet the expectations of their professional or social networks.

4. *Integrating personalized feedback and support*
   Providing personalized, constructive feedback is crucial for addressing learners' specific

needs, challenges, and progress toward their goals. Instructors can offer feedback that is both encouraging and practical, helping learners understand how their current language proficiency supports their professional or personal aspirations. For example, feedback like, "You've done a great job using technical terms in your presentation. With a bit more work on your pronunciation, you'll feel even more confident in client meetings," highlights both strengths and areas for improvement. Personalized feedback demonstrates that the instructor understands the learner's goals and expectations, which can significantly enhance the learner's motivation to work toward their ought-to L2 self. It also reinforces the idea that the course is not just about language acquisition, but about achieving tangible, practical results in their daily lives.

5. *Creating a flexible learning environment*
   Offering a flexible learning structure is essential for adult learners who need to balance their studies with work and other responsibilities. To support this, instructors can provide asynchronous learning options, such as videos, readings, and self-paced assignments, allowing learners to access content on their own schedules. Additionally, offering synchronous sessions at varying times accommodates different time zones or work commitments. This flexibility helps adult learners stay motivated without feeling overwhelmed, enabling them to meet both their professional development or personal goals and their other responsibilities. By catering to their ought-to L2 self, this approach ensures that learners can fulfill external expectations while maintaining a healthy work-life balance.

## 9. Conclusions

This sequential mixed-methods study delved into the complex interrelationships among L2 self-guides, enjoyment, and engagement in the online context among Chinese L2 learners. The quantitative phase yielded intriguing results, demonstrating that the ideal L2 self did not exert a significant influence on engagement and enjoyment. Conversely, ought-to L2 emerged as a significant positive predictor for both engagement and enjoyment. Furthermore, it was revealed that ought-to L2 self could also indirectly affect engagement through the mediating role of enjoyment. The qualitative strand of this study served to validate and elucidate the quantitative findings, offering a deeper understanding of the observed statistical patterns. Collectively, these findings highlight the intricate motivational dynamics at play in the online L2 learning milieu and suggest that a more granular examination of the social-economic factors that stimulate engagement and enjoyment among adult learners is both necessary and warranted.

This study has several inherent limitations that require explicit acknowledgment. Firstly, the cross-sectional design of the study imposes constraints on establishing causal relationships between L2 self-guides, enjoyment, and engagement among Chinese adult L2 learners in online learning contexts over an extended period. To address this issue, future research could adopt a cross-lagged panel design, allowing for the examination of longitudinal causality among these variables [78]. Additionally, financial constraints limited our study's sample size and scope. The study included only 367 participants from three Chinese universities, and the qualitative phase involved just five interviewees. To mitigate this limitation, future research should expand recruitment efforts to include a more diverse array of universities and interview subjects. By broadening the sample size and diversity, future studies can enhance both the generalizability of their findings and the depth of understanding of the varied experiences of adult L2 learners in online educational settings.

## Supporting information

**S1 File. Original data.**
(ZIP)

## Author Contributions

**Conceptualization:** Yanyan Li, Wentao Liu, Shuai Ren.

**Data curation:** Yanyan Li, Wentao Liu, Shuai Ren.

**Formal analysis:** Yanyan Li, Wentao Liu, Shuai Ren.

**Funding acquisition:** Wentao Liu, Shuai Ren.

**Investigation:** Yanyan Li, Wentao Liu, Shuai Ren.

**Methodology:** Yanyan Li, Wentao Liu, Shuai Ren.

**Project administration:** Wentao Liu, Shuai Ren.

**Resources:** Yanyan Li, Wentao Liu.

**Software:** Yanyan Li, Wentao Liu, Shuai Ren.

**Supervision:** Wentao Liu.

**Validation:** Yanyan Li, Wentao Liu, Shuai Ren.

**Visualization:** Yanyan Li, Wentao Liu.

**Writing – original draft:** Yanyan Li, Wentao Liu.

**Writing – review & editing:** Yanyan Li, Wentao Liu, Shuai Ren.

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
