## [Decision Letter · Decision Letter 0]

21 Nov 2024

PONE-D-24-44890What sustain Chinese adult second language (L2) learners' engagement in online classes? A sequential mix-methods study on the roles of L2 motivation and enjoymentPLOS ONE

Dear Dr. Liu,

Thank you for submitting your manuscript to PLOS ONE. After careful consideration, we feel that it has merit but does not fully meet PLOS ONE’s publication criteria as it currently stands. Therefore, we invite you to submit a revised version of the manuscript that addresses the points raised during the review process.

We look forward to receiving your revised manuscript.

Kind regards,

Hua Pang

Academic Editor

PLOS ONE

Journal Requirements:

2. We note that your Data Availability Statement is currently as follows: [All relevant data are within the manuscript and its Supporting Information files.] Please confirm at this time whether or not your submission contains all raw data required to replicate the results of your study. Authors must share the “minimal data set” for their submission. PLOS defines the minimal data set to consist of the data required to replicate all study findings reported in the article, as well as related metadata and methods (https://journals.plos.org/plosone/s/data-availability#loc-minimal-data-set-definition). For example, authors should submit the following data: - The values behind the means, standard deviations and other measures reported; - The values used to build graphs; - The points extracted from images for analysis. Authors do not need to submit their entire data set if only a portion of the data was used in the reported study. If your submission does not contain these data, please either upload them as Supporting Information files or deposit them to a stable, public repository and provide us with the relevant URLs, DOIs, or accession numbers. For a list of recommended repositories, please see https://journals.plos.org/plosone/s/recommended-repositories. If there are ethical or legal restrictions on sharing a de-identified data set, please explain them in detail (e.g., data contain potentially sensitive information, data are owned by a third-party organization, etc.) and who has imposed them (e.g., an ethics committee). Please also provide contact information for a data access committee, ethics committee, or other institutional body to which data requests may be sent. If data are owned by a third party, please indicate how others may request data access.

Reviewers' comments:

Reviewer's Responses to Questions

**Comments to the Author**

1. Is the manuscript technically sound, and do the data support the conclusions?

Reviewer #1: Partly

Reviewer #2: Yes

2. Has the statistical analysis been performed appropriately and rigorously? 

Reviewer #1: Yes

Reviewer #2: Yes

3. Have the authors made all data underlying the findings in their manuscript fully available?

Reviewer #1: Yes

Reviewer #2: Yes

4. Is the manuscript presented in an intelligible fashion and written in standard English?

Reviewer #1: No

Reviewer #2: Yes

5. Review Comments to the Author

Reviewer #1: The paper's topic has certain theoretical and practical significance. The overall design is well-executed, aligning with the paradigm and approach of mixed research design. The structure of the paper is complete, and the quantitative and qualitative data effectively address the research questions. However, there are several issues that the authors should consider:

[1] The grasp of prior literature for this research needs to be strengthened. Please consider the following references

Liu, H., Zhong, Y., Chen, H., & Wang, Y. (2023). The mediating roles of resilience and motivation in the relationship between students’ English learning burnout on engagement: A conservation-of-resources perspective. International Review of Applied Linguistics in Language Teaching. https://doi.org/10.1515/iral-2023-0089.

Wang, X. & Liu, H. (2024). Exploring the moderation roles of emotions, attitudes, environment, and teachers on the impact of motivation on learning behaviors in students’ English learning. Psychological Reports, online publication, 1-27. https://doi.org/10.1177/00332941241231714

Wang, Y. & Liu. H. (2022). The mediating roles of boredom and buoyancy in the relationship between autonomous motivation and engagement among Chinese senior high school EFL learners. Frontiers in Psychology. 13:992279, 1-12. https://doi.org/10.3389/fpsyg.2022.992279

Liu, H. & Fan, J. (2024), AI-mediated communication in EFL classrooms: the role of technical and pedagogical stimuli and the mediating effects of AI literacy and enjoyment. European Journal of Education. e12813. https://doi.org/10.1111/ejed.12813

Liu, H., Zhu, Z., & Chen, B. (2024). Unraveling the mediating role of buoyancy in the relationship between anxiety and EFL students’ learning engagement. Perceptual and Motor Skills. https://doi.org/10.1177/00315125241291639

Liu, H., Li. J., & Fang. F. (2022). Examining the complexity between boredom and engagement in English Learning: Evidence from Chinese high school students. Sustainability, 14, 16920: 1-12. https://doi.org/10.3390/su142416920.

Liu, H. & Song, X. (2021). Exploring “Flow” in young Chinese EFL learners’ online English learning activities. SYSTEM, 96, 102425: 1-13. https://doi.org/10.1016/j.system.2020.102425.

Please note you are not obliged to include any of the citations above. Not including the suggested

references has no negative impact on the final editorial decision. You are requested to determine

if the citations add value to the current submission and are free to exclude any/all citations as

appropriate.

[2] Figures should use standardized models instead of non-standardized ones. Additionally, some correlation coefficients are not clear from the figures, such as the relationship between IL2S and several other variables.

[3] The authors are requested to check whether the factor loading from engagement to cognition being 1.01 is abnormal, which leads back to the question above about whether non-standardized figures were used.

[4] The text throughout the paper requires polishing and proper copyediting.

Reviewer #2: This study addresses an important and under-researched area - the motivational and emotional factors influencing adult L2 learners' engagement in online language classes. The authors employ a robust theoretical framework, the Control-Value Theory, to examine the interplay between L2 self-guides (ideal L2 self and ought-to L2 self), enjoyment, and engagement among Chinese adult L2 learners. The sequential mixed-methods design allows for both breadth and depth in the exploration of these relationships. While the study has several strengths, there are also areas that require further clarification and development before the manuscript is ready for publication.

One of the key strengths of this study is its focus on adult L2 learners, a demographic that has received limited attention in the existing literature compared to school-aged learners. Given the distinct challenges and motivations of adult learners, such as balancing language learning with professional and personal obligations, this study fills an important gap in our understanding. The authors make a compelling case for why this population warrants special consideration in L2 motivation research.

The theoretical grounding of the study in the Control-Value Theory is another strength. The authors provide a clear rationale for how the CVT framework can elucidate the relationships between L2 self-guides, enjoyment, and engagement. The visual representation of the theoretical links (Figure 1) effectively illustrates the proposed pathways. Situating the study within this established theory enhances the conceptual clarity and interpretability of the findings.

Methodologically, the use of a sequential mixed-methods design is well-justified. The quantitative phase, with its large sample size (n=367), allows for the statistical testing of the hypothesized relationships using structural equation modeling. The subsequent qualitative phase, involving semi-structured interviews with a subset of participants, provides a deeper understanding of the statistical patterns observed. This integration of quantitative and qualitative data strengthens the validity and richness of the conclusions drawn.

However, there are several areas where the manuscript could be improved. First, while the authors mention that the scales used to measure L2 self-guides, enjoyment, and engagement were adapted from previous studies, more details are needed about this adaptation process. How exactly were the items modified to fit the context of adult online learners? Were any new items developed? Providing this information would enhance the replicability of the study.

Second, the description of the qualitative data analysis procedures is somewhat sparse. The authors state that they followed the six-phase thematic analysis approach outlined by Braun and Clarke (2006), but do not provide much detail on how each phase was implemented. For example, how were initial codes generated? How were themes reviewed and refined? A more transparent and thorough account of the qualitative analysis process would strengthen the trustworthiness of the findings.

Third, the discussion of the quantitative results could be more fully developed. While the authors do an excellent job of presenting the SEM findings and relating them back to the hypotheses, there is limited exploration of why certain hypotheses were not supported. For instance, why might the ideal L2 self not significantly predict enjoyment and engagement in this context, contrary to previous studies? The authors offer some potential explanations based on the qualitative data, but a deeper engagement with the extant literature and theory would enrich the interpretation.

Finally, the practical implications of the study could be more clearly articulated. Given the findings about the importance of the ought-to L2 self for adult learners' enjoyment and engagement, what specific strategies might online language programs employ to leverage this? How can course designers and instructors create environments that cater to the unique motivations of adult learners? Providing more concrete recommendations grounded in the study's insights would enhance the applied value of the research.

In terms of minor issues, there are a few instances of awkward phrasing or grammatical errors (e.g., "Our study collected quantitative data during summer from three universities in China that offer multilingual online training for adult L2 learners" could be rephrased for clarity). A thorough round of proofreading and editing would help polish the writing.

Overall, this study makes a valuable contribution to our understanding of the motivational dynamics underlying adult L2 learners' engagement in online classes. The focus on an under-researched population, the solid theoretical foundation, and the mixed-methods approach are notable strengths. However, the manuscript would benefit from more methodological detail, a deeper discussion of the findings in light of prior research, and clearer practical implications. With some revisions to address these points, the paper has the potential to make a significant impact in the field of L2 motivation.

6. PLOS authors have the option to publish the peer review history of their article (what does this mean?). If published, this will include your full peer review and any attached files.

Reviewer #1: No

Reviewer #2: No

---

## [Author Response · Author response to Decision Letter 0]

18 Dec 2024

Dear Reviewer 1 and 2,

We would like to express our sincere gratitude for your expertise and generosity in reviewing our manuscript. Your insightful comments have significantly contributed to enhancing the rigor and depth of our work, and we deeply appreciate your valuable input. Kindly find our revised manuscript and the accompanying Response to Reviewers, where we have detailed the revisions made in response to the feedback.

Best wishes

---

## [Decision Letter · Decision Letter 1]

5 Jan 2025

What sustain Chinese adult second language (L2) learners' engagement in online classes? A sequential mix-methods study on the roles of L2 motivation and enjoyment

PONE-D-24-44890R1

Dear Dr. Liu,

We’re pleased to inform you that your manuscript has been judged scientifically suitable for publication and will be formally accepted for publication once it meets all outstanding technical requirements.

Kind regards,

Hua Pang

Academic Editor

PLOS ONE

Additional Editor Comments (optional):

Reviewers' comments:

Reviewer's Responses to Questions

**Comments to the Author**

1. If the authors have adequately addressed your comments raised in a previous round of review and you feel that this manuscript is now acceptable for publication, you may indicate that here to bypass the “Comments to the Author” section, enter your conflict of interest statement in the “Confidential to Editor” section, and submit your "Accept" recommendation.

Reviewer #1: All comments have been addressed

Reviewer #2: (No Response)

2. Is the manuscript technically sound, and do the data support the conclusions?

Reviewer #1: Yes

Reviewer #2: (No Response)

3. Has the statistical analysis been performed appropriately and rigorously? 

Reviewer #1: Yes

Reviewer #2: (No Response)

4. Have the authors made all data underlying the findings in their manuscript fully available?

Reviewer #1: Yes

Reviewer #2: (No Response)

5. Is the manuscript presented in an intelligible fashion and written in standard English?

Reviewer #1: Yes

Reviewer #2: (No Response)

6. Review Comments to the Author

Reviewer #1: (No Response)

Reviewer #2: Thanks for the revisions. The revised file looks fine to me. The only minor comment is that the author should carefully proofread the paper for better readability.

7. PLOS authors have the option to publish the peer review history of their article (what does this mean?). If published, this will include your full peer review and any attached files.

Reviewer #1: No

Reviewer #2: No

---

## [Editor Report · Acceptance letter]

13 Jan 2025

PONE-D-24-44890R1 

PLOS ONE

Dear Dr. Liu, 

I'm pleased to inform you that your manuscript has been deemed suitable for publication in PLOS ONE. Congratulations! Your manuscript is now being handed over to our production team.

Kind regards, 

on behalf of

Dr. Hua Pang 

Academic Editor

PLOS ONE